# Statistical Parametric Mapping Reveals Subtle Gender Differences in Angular Movements in Table Tennis Topspin Backhand

**DOI:** 10.3390/ijerph17196996

**Published:** 2020-09-24

**Authors:** Ziemowit Bańkosz, Sławomir Winiarski

**Affiliations:** Department of Biomechanics, Faculty of Physical Education and Sports, University School of Physical, Education in Wrocław, al. Paderewskiego 35, 51-612 Wrocław, Poland; slawomir.winiarski@awf.wroc.pl

**Keywords:** statistical parametric mapping, gender differences, movement pattern, kinematics, table tennis, sports technique

## Abstract

*Background:* Statistical parametric mapping (SPM) is an innovative method based on the analysis of time series (data series) and is equivalent to statistical methods for numerical (discrete) data series. This study aimed to analyze the patterns of movement in the topspin backhand stroke in table tennis and to use SPM to compare these patterns between advanced female and male players. *Methods:* The research involved seven advanced male and six advanced female players. The kinematic parameters were measured using an inertial motion analysis system. The SPM was computed using the SPM1D Python package. *Results:* Our study made it possible to reproduce the pattern of movement in the joints during topspin backhand strokes in the studied athletes. During multiple comparisons, the analysis of variance (ANOVA) SPM test revealed many areas in the studied parameter series with statistically significant differences (*p* ≤ 0.01). *Conclusions:* The study presents the movement patterns in the topspin backhand shot and describes the proximal-to-distal sequencing principle during this shot. The SPM study revealed differences between men and women in the contribution of thoracic rotation, external shoulder rotation, dorsal flexion, and supination in the wrist during the hitting phase. These differences may result from the anatomical gender differences or variations in other functionalities of individual body segments between the study groups. Another possible source for these discrepancies may reside in tactical requirements, especially the need for a more vigorous attack in men. The gender differences presented in this study can help in the individualization of the training process in table tennis.

## 1. Introduction

### 1.1. Table Tennis Kinematics in Scientific Literature

Table tennis is a very complex and comprehensive sport. The technique and tactics of playing, mental and physical capacity, and fitness are some of the components that affect the achievement of a champion level in the sport. The technique of playing table tennis includes many elements, such as footwork (small steps, sidestep, cross-over step, single step, pivot, etc.), single strokes (topspin forehand and backhand, smash, service, flip, block, and others), and combining strokes (such as linking topspin backhand and forehand, serve and attack, receive and counterattack, or long serve and counterattack) [1,2]. Each stroke can be determined by the speed (such as the speed of stroke movement, the moment of impact, time of reaction, and taking action), the impact force (the value of the force directed toward the flying ball), rotation (the manifestation of force directed tangentially to the ball or the effect of rubbing the ball with the racket), or the ball placement on the table. The kinematic analysis of table tennis strokes is an increasingly common issue described in the literature. Most often, researchers in the field of table tennis evaluate the topspin stroke, as it is considered the most commonly used one to score a point [3,4,5,6]. For example, authors dealing with the kinematics of topspin strokes compared athletes at different sports skill levels [5,7,8,9] and checked for differences between individual strokes [5,10]. Other authors searched for correlations between the kinematics of selected body segments and racket speed [6,11]. Changes in topspin kinematics caused by psychological pressure were also evaluated [12].

### 1.2. Variability—An Interesting Issue in Table Tennis

Table tennis is a sport with an almost infinite number of player behaviors and movement possibilities (such as strokes, positions, and tactical variations); therefore, variability—such as behavioral, mental and physical—in this sport seems to be an appealing direction. Few studies in the literature on table tennis kinematics have evaluated the variability of movement. This variability is increasingly being considered in scientific studies and explained using a number of theories. For example, according to the generalized motor program theory (GMPT), the variability of movement results from errors (errors of prediction, evaluation, or choice of movement parameters) and is reduced by repetition and learning of a specific activity [13,14]. Using the uncontrolled manifold (UCM) approach, other authors present a theory according to which the variability of movement results from the excess of options of movement responses (degrees of freedom), which results in a large possibility of solutions for the same task [15]. According to these authors, the variance in multi-degree-of-freedom systems is decomposed into a variance that leaves task variables invariant (UCM) and one that does not (orthogonal to the UCM). A larger variance within the UCM than orthogonal to it is interpreted as evidence for a task-specific solution of the multi-degree-of-freedom problem. The theory of dynamic systems used in sports suggests that a high variability of movement is attributed to the greater capabilities (e.g., skills) of an athlete. Many authors perceive the variability of movement as functional variability, a natural phenomenon that is an expression of the functional adjustment of movement to the requirements of the environment or the task, of individualization, and of the reduction of the risk of injury [14,16]. Assessing the occurrence and scale of variability of movement appears to be extremely important in the process of teaching and improving the skills of purposive movements and explaining how to control human movements. Evaluation of the magnitude, significance, and interpretation of the variability of movement depends on the method used [14]. For example, analyses of the magnitude of the standard deviation and the coefficient of variation as linear measures are used to evaluate this variability. Other tools and methods used include approximate entropy, sample entropy, correlation dimension, largest Lyapunov exponent, and detrended fluctuation analysis [14].

### 1.3. Statistical Parametric Mapping

Statistical parametric mapping (SPM) is the application of random field theory. It is the gold standard statistical method dedicated to numerical signal data analysis. SPM is equivalent to the statistical methods for discrete data series. It was developed at the Wellcome Department of Imaging Neuroscience (University College London, London, UK) by Karl Friston to analyze the differences in brain activity recorded during neuroimaging (Functional magnetic resonance imaging—fMRI, Positron emission tomography—PET, Single-photon emission computed tomography—SPECT, Electroencephalography - EEG or Magnetoencephalography - MEG imaging). The SPM model is used for each voxel (approximately 3 × 3 × 3 mm spatial image unit) using a general linear model to describe the variability of data in terms of experimental and residual variability effects. The hypotheses expressed in categories of model parameters are evaluated for each voxel using one-way analysis [17]. For the one-dimensional variables recorded with the motion analysis system, the general SPM model can be simplified to the one-dimensional model spm.1d. Recently, the one-dimensional model has been implemented in the Matlab and Python calculation algorithms (www.spm1d.org) and used for comparisons of time waveforms for kinematic and dynamic variables [18,19].

Recent advances in SPM for continuum one-dimensional data (e.g., kinematic or kinetic time series) have been adopted by the research community—e.g., scientists, clinicians, engineers, and all individuals who deal with spatiotemporally continuous data. Originally adopted for the analysis of pedobarographic images [20], SPM has also been useful in uni- or multivariate time series analysis (kinematics, kinetics, or surface Electromyography—EMG data) [18,19,21]. Recently, a classical SPM supplemented with a Bayesian version was proved to be more sensitive to assess significant side differences, is not asymmetric like the classical version, and takes an alternative hypothesis into account [22]. The SPM is especially suitable for biomechanical data that is spatiotemporally smooth, sampled above the Nyquist frequency, and bounded in time or space [23]. The two main advantages of SPM over other numerical approaches (such as the autocorrelation/autocovariance function, moving average or autoregressive representations of time series processes, exponential smoothing, Fourier analysis, or time/vector time series analysis) are: (1) the statistical results are presented directly in the original sampling space, so their spatiotemporal biomechanical framework is immediately apparent and (2) there is no need for assumptions regarding the spatiotemporal central points of signals [19]. As other methods, SPM also has some known limitations: e.g., it requires temporal data normalization, non-random sampling, and non-blind experimentation; its procedures are more complex—the random-field-theory-based inference especially; and it presents restricted accessibility (to Python/Matlab programming).

### 1.4. Research Goals Motivation

The few available studies on table tennis and the variability of movement have been based on the methods of evaluation of standard deviation, correlation, and analysis of variance (ANOVA and Least Significant Difference—LSD) and presented UCM calculations. In general, these works evaluated the mechanism of controlling the forehand topspin movement by observing the variability of its parameters in the initial phase and at the moment of contact between the racket and the ball [24,25]. Iino, Yioshioka, and Fukashimo emphasized that the possibility of using different configurations in the evaluated joints to stabilize the vertical angle of the racket in table tennis strokes can be a critical factor in playing performance [3]. A previous study of Bańkosz and Winiarski also evaluated the variability of movement by analyzing the coefficient of variation of kinematic parameters in selected important moments of the hitting movement [26]. However, the coordination of movements in individual joints was not taken into account. The variability of temporal and spatial coordination of movements, the possibility of compensation, and functional variability are significant problems in the coaching practice and in the process of teaching and improving technique and its monitoring [3]. Making the coaches and players aware of the different variants of strokes even for a specific solution (e.g., playing with the right strength, speed, and rotation to the same place) seems to be very important and necessary for improving the training process. Taking into account the differences between athletes and looking for individual technical solutions is, therefore, a better choice than, for example, copying and imposing a single pattern of performing the movement. It seems interesting to answer the question of what is the range of sex differences and similarities in the performance of topspin strokes during the movement in the most important joints. A recent study of Bańkosz, Winiarski, and Malagoli Lanzoni evaluated differences between the angular and kinematic parameters in the most important events, but the analysis of movement coordination was based only on the magnitude of joint angles in these events [27]. Therefore, the study aimed to analyze the patterns of movement in joints selected as the most important during a topspin backhand stroke and to use SPM to compare these patterns between advanced female and male table tennis players. Based on previous studies, we proposed the hypothesis that gender differences would be revealed in the movement patterns of segments of the upper body and distal segments of the hand.

## 2. Materials and Methods

This study involved seven male and six female table tennis players at an international sport skill level (national team) aged 21.8 ± 3.1 y and 20.2 ± 1.4 y, respectively, with an average body height of 177.3 ± 2.5 and 166 ± 1.8 cm, respectively, body mass of 76 ± 6.5 and 57 ± 3.9 kg, respectively, and years of experience 15.0 ± 1.9 and 13.2 ± 1.2, respectively. Before the start of the study, all participants were informed about the purpose of the study and the possibility of withdrawing participation at any stage and without giving a reason. All procedures performed in this study were in accordance with the ethical standards of the Helsinki Declaration. The research project received a positive opinion from the Senate’s Research Bioethics Commission at the University School of Physical Education in Wrocław, Poland (Ethics IRB number 34/2019).

The kinematic parameters were measured using an MR3 myoMuscle Master Edition system (myoMOTION™, Noraxon, Scottsdale, AZ, USA—Figure 1). The myoMOTION system consists of a set of 1 to 16 sensors using inertial sensor technology. Based on the so-called fusion algorithms, the information from a 3D accelerometer, gyroscope, and magnetometer is used to measure the 3D rotation angles of each sensor in absolute space (yaw-pitch-roll, also called orientation or navigation angles). The inertial sensors were located on the body of the study participant to record the accelerations according to the myoMotion protocol, described in the instruction manual (Figure 1). The accuracy and validity of the inertial measurement units (IMU) system in angle determination is unquestioned and was the subject of previous research [28].

The participants performed the task of a topspin backhand (TBH) as a response to a backspin ball, repeated 15 times. Sensors were attached with elastic straps and self-adhesive tape symmetrically so that the y-coordinate corresponded to the frontal horizontal axis, and the z-coordinate to the sagittal horizontal axis of the segment (Figure 1). Positive vertical x-coordinate on the sensor label corresponded to a superior orientation for the trunk, head, and pelvis (Figure 1). For the limb segment sensors, the positive x-coordinate corresponded to a proximal orientation. For the foot sensor, the x-coordinate was directed distally (to the toes). The sensors were placed according to the myoMotion protocol described in the manual. At the beginning of the measurement, each participant was checked and the system was calibrated according to the factory recommendations. The maximal sampling rate for a given sensor/receiver was 100 Hz per sensor for the whole 16-sensor set and was adjusted to the recording speed by the piezoelectric sensor (1500 Hz). Noraxon’s IMU technology mathematically combines and filters the incoming source signals at the sensor level and transmits the four quaternions of each sensor. We used system-build fusion algorithms and Kalman filtering—a digital bandpass finite impulse response (FIR) filter. This mode allowed direct access to all unprocessed IMU sensor data.

The task was composed of 15 specified strokes, and the player was asked to hit the marked area in the corner of the table (30 × 30 cm) diagonally (the exact instruction was, “Play diagonally, accurately, and as hard as you can”). After the video analysis, only successful shots considered “on table” and played diagonally were recorded for further calculations (missed balls, balls hit out of bounds, and balls hit into the net were excluded). The balls were shot by a dedicated table tennis robot (Nevgy Robo Pong Robot 2050, Nevgy Industries, Hendersonville, TN, US —Figure 1) at constant parameters of rotation, speed, direction, and flight trajectory. The settings of the robot were as follows:
rotation type: backspinspeed (determines both speed and spin, where 0 is the minimum, and 30 is the maximum): 11left position (leftmost position to which the ball is delivered): 15wing (robot’s head angle indicator): 9.5frequency (time interval between balls thrown): 1.4 s

For the experiment, we used plastic Andro Speedball 3S 40+ balls (Andro, Dortmund, Germany), a Stiga Premium Compact table (Stiga, Eskilstuna, Sweden), and the same racket with the following characteristics: blade—Jonyer-H-AN (Butterfly Tokyo, Japan) and rubber—Tenergy 05, 2.1 mm (Butterfly, Japan).

Following the International Society of Biomechanics (ISB) recommendations on definitions of the joint coordinate system of various joints for the reporting of human joint motion [29,30], the following angles were chosen for both sides and sampled every 0.01% of cycle time:Knee flexion-extension (KFE): movement of the tibia with respect to the femur coordinate system in the sagittal plane due to the rotation of the proximal–distal axis about the mediolateral axis; a negative sign denotes extension, and positive sign flexion;Hip flexion-extension (HFE): movement of the femur with respect to the pelvis coordinate system in the sagittal plane due to the rotation of the proximal–distal axis about the mediolateral axis; a negative sign denotes extension, while positive flexion;Hip abduction-adduction (HAA): movement of the femur with respect to the pelvis coordinate system in the frontal plane due to the rotation of the proximal–distal axis out of the sagittal plane; a negative sign denotes adduction, while positive abduction;Hip internal-external rotation (HIER): internal or external movement of the femur with respect to the pelvis coordinate system in the transversal plane due to the rotation about the proximal–distal axis; a negative sign denotes internal, while positive external rotation;Lumbar internal-external rotation (LIER): internal or external movement of the loins in the transversal plane due to the rotation about the loin longitudinal axis; a negative sign denotes internal, while positive external rotation;Thoracic internal-external rotation (ThIER): internal or external movement of the thorax relative to the global coordination system in the transversal plane due to the rotation about the thorax longitudinal axis; a negative sign denotes internal, while positive external rotation.

For the upper extremity (playing side), a simplified biomechanical model was adopted based on the predominant plane of movement as described by Kontaxis et al. [31] with the following segments of interest: thorax, clavicle, scapula, humerus, forearm, and carpus of the hand. The joints of interest were the sternoclavicular/girdle, scapulothoracic, glenohumeral, elbow, and wrist. Based on the adopted sequence of Euler angles, the following angles were computed:Shoulder flexion-extension (ShFE): movement of the humerus relative to the thorax in the sagittal plane; a negative sign denotes extension, while positive flexion;Shoulder abduction-adduction (ShAA): movement of the humerus relative to the thorax in the frontal plane; a negative sign denotes adduction, while positive abduction;Shoulder internal-external rotation (ShIER): movement of the humerus relative to the thorax in the transversal plane; a negative sign denotes internal (medial), while positive external (lateral) rotation;Elbow flexion-extension (EFE): movement of the forearm relative to the humerus along the transversal axis; a negative sign denotes (hyper)extension, while positive flexion;Wrist flexion-extension (WFE): movement of the wrist relative to the radius along the transversal axis and measured between the upper arm and hand sensors; a negative sign denotes extension, while positive flexion;Wrist supination-pronation (WSup): movement of the wrist relative to the radius along the axis and measured between the upper arm and hand sensors; pronation is a positive rotation, while supination is a negative rotation;Wrist radial abduction-adduction (WRad): movement of the wrist relative to the radius and measured between the upper arm and hand sensors; adduction (or ulnar deviation) is negative, while abduction (or ulnar deviation) is positive.

The movement of the playing hand was used to assess specific events of the cycle (Figure 2):Ready position: the hand is not moving after the previous stroke, just before the swing;Backswing: the moment when the hand changes direction from backward to forward in the sagittal plane after the swing;ACCMax: the moment when the hand reaches the maximum acceleration;Forward: the moment when the hand changes direction from forward to backward in the sagittal plane after the stroke (the end of the cycle and the beginning of next cycle).

The phases between the defined events were as follows: the back to ready position phase (between the forward and ready positions), backswing phase (between the ready and backswing positions), hitting phase (between backswing and ACCMax) and forward end phase (between ACCMax and forward).

The statistical calculations were performed using Statistica 13.1 (TIBCO Software Inc). The Shapiro–Wilk test was used to assess the normality of the data distributions. The descriptive statistics were calculated (mean, SD). The SPM was calculated using the SPM1D Python package, which offers a high-level interface to 1D SPM. The angle-time numerical series were averaged over trials and reported against cycle time (Figure 3A). For each participant and selected time-dependent angular numerical data, a two-sample t-test—the SPM{t} function (with alpha = 0.05, non-sphericity correction, and assumption of unequal variances)—was numerically computed to check the level of similarity between the movements [30]. For each test, a statistical parametric map, SPM{t} (Figure 3B), was created by calculating the conventional univariate t-statistic at each point of the gait curve [30]. When an SPM{t} crossed the assumed threshold, an additional threshold cluster was created, indicating a significant difference (a gray area) between two compared joint motion patterns in a specific location of the gait cycle. In the present study, because of the high number of statistical analyses, the SPM results are visualized in a summarized manner. Instead of SPM{t} curves, blue bars indicate the significance during the gait cycle in Figure 3C.

## 3. Results

The data obtained in the study (all the data is available in Appendix A) made it possible to reproduce the pattern of movement in the joints during topspin backhand strokes in the athletes studied. The pattern is presented in Figure 4, Figure 5 and Figure 6 where the horizontal lines represent the mean ± standard deviation (SD), the vertical lines are the individual identified events, and the fields at the bottom show the ANOVA SPM(t) test results.

It can be observed that the whole phase of hitting is longer in men and the times of it beginning and ending vary statistically (Table 1).

During the comparison of the movement patterns in individual joints in women and men, the ANOVA SPM(t) test showed many periods with significant differences. For the knee joints on the playing side, the movement pattern in both groups was mostly different. Although a flexion is observed in both groups in the backswing phase, an extension in the hitting phase (already after its beginning), with different angles in both groups was observed (Figure 4). No differences occur during around 20% of the cycle time and just after the beginning of the backswing phase. Similar differences occur for the knee joints on the non-playing side (Figure 4), and the extension movement at these joints occurs just before the beginning of the hitting phase. At the hip joints on the playing side, the ANOVA test showed fewer periods of differentiation between the two groups. They occur during ca. 5–10% of the cycle (in the back to ready position phase, in a significant part of the backswing phase, and during the second half of the hitting phase). However, flexion can be noticed in both groups in the backswing phase, whereas extension occurred before the hitting to forward phase (Figure 4). Likewise, many periods without differentiation between the groups were observed at the hip joints on the non-playing side mainly during the backswing and hitting phases, whereas extension in these joints—which supports the hitting movement—starts before the hitting phase (Figure 4). There are significant differences in the use of rotation in the hip joint, but large and higher SD values in men can be observed compared to women. Differences were detected in the second part of the backswing phase and the whole hitting phase (Figure 4) for abduction at the hip joint on the playing side, first with noticeable abduction in both groups (in the backswing phase), followed by adduction at around the middle of the hitting phase (Figure 4). Furthermore, at the hip joint on the non-playing side, the differences between the groups were observed for approximately the first 55% of the cycle time. In the backswing phase, this is a small adduction movement, whereas, after the beginning of the hitting phase, abduction occurs at a large SD in men.

Differences in thoracic rotation can be observed in the second part of the backswing phase and in most of the hitting phase, from ca. 15% to 65% of the cycle (Figure 5). In men, this segment rotates to the left in the backswing phase, and when the hitting phase begins, this movement starts to the right, a phenomenon not seen in women (Figure 5).

For the shoulder flexion, several significantly different periods of movement are noticeable between the two groups, especially during the hitting phase (Figure 6). Although the flexion movement can be observed in both groups, it is faster in men (the shoulder flexion range is bigger in males by ca. 20 deg. than in females in the hitting phase), ending immediately after reaching ACCMax, while in women, it is slower and ends later—in fact, when the forward phase starts (Figure 6). Similar differences can be observed in shoulder abduction, although there are more periods with significant differences: in the hitting phase in men, abduction is faster than in women and ends at ACCMax and the range of movement is also more significant: ca. 100 deg. in men compared to 50 deg. in women. The shoulder rotation movement shows differences between the groups not only at the beginning of the forward end phase but over almost the entire striking cycle. This movement is similar in both groups, but the internal rotation is slightly higher in men, while the external rotation is lower in this group. The ACCMax event in men occurs at ca. 75 deg. of internal rotation, whereas in women, at ca. 35 deg. (Figure 6). Many differences between the groups can be observed for the EFE movement. The range of this movement in women is higher than in men, whereas flexion in the back to ready position, although noticeable in both groups, is faster and deeper in women. The extension movement, which supports the stroke, begins in the backswing phase for women, while for men, it begins in the middle of the hitting phase. In the wrist joints, differences in the flexion/extension movement occur in the vast majority of the movement (Figure 6). In the backswing phase, the flexion movement can be observed to start at ca. 5 deg. in mean and at ca. 25 deg. in women. At the end of the hitting phase, both groups (after reaching ca. 35 deg. of flexion) change their direction of movement to extension (this is the moment when the SPM shows no differences). In men, the movement of extension is faster and with a higher range than in women. Similar differences occur in the radial adduction/abduction movement of the wrist during the major part of the movement in the joint (except for the hitting phase). In the backswing phase, this movement is a radial abduction in a small range, which increases faster in the hitting phase compared to the previous phase. In about half of the hitting phase, the movement changes direction to the radial adduction, and the SPM test within this change does not indicate differences between the groups (Figure 6). The wrist pronation/supination movement shows more periods without any differences between the groups: no differences occur during the back to ready position phase, at the beginning of the backswing phase and, as in previous cases, around the middle of the hitting phase (when the direction of the movement changes from pronation to supination) (Figure 6). The manifestation of the principle of proximal-to-distal sequencing movement coordination can be observed in both groups, with the directions of movements in the limb and body trunk joints changing first and the wrist joint added as the last into the hitting movement (in the middle of the hitting phase or later) (Figure 6).

## 4. Discussion

Our study aimed to analyze the patterns of movement in selected joints during a topspin backhand stroke and to use the SPM method to compare these patterns between advanced female and male table tennis players. Many literature sources indicate the principle of the proximal-to-distal sequence is used for throwing and striking movements, defining the proper technique [32,33]. According to this principle, such movements are initiated within the proximal central lines of the body (e.g., lower limbs, pelvic girdle, and upper body in throwing) and, with a specific time shift, ended with the movement of distal parts—i.e., the hand. In table tennis, this principle and its specific use have not been sufficiently described to date. The existence of this principle in the coordination of topspin strokes was emphasized in previous studies [34]. In the present work, both in the group of men and women, this principle is reflected in the moments of change in the direction and speed of movement (i.e., range of movement in the phase) in the backswing, hitting, and forward end phases, as shown in the figures. In both groups studied, during a topspin backhand stroke against the backspin ball, flexion is observed in the knee and hip joints during in the backswing phase, extension from the position of the playing limb close to the body (ca. 0 deg. of flexion/extension and 0 deg. of adduction/abduction), internal rotation (25–50 deg.) in the shoulder joint, flexion in the elbow joint (60–80 deg.), and positioning in the wrist joint of the hand in abduction, pronation, and flexion. The change in direction and increase in speed are observed first (before the beginning of the hitting phase) in the knee joints (extension, first in the legs on the non-playing side), in the shoulder flexion, and external rotation movement. Based on the movements described in other sports [35,36], before the start of the hitting phase, one could also expect a movement of the body trunk rotation (spine in the lumbar and thoracic segments). This movement, observed in women to a small extent, while in men with a slightly more pronounced range (a dozen or so degrees), begins just after this phase has started. This phenomenon indicates a small contribution of this movement when hitting topspin backhand, especially in women. The elbow extension movement is activated in the middle (or even a little later) of the hitting phase. In the group of women, the movement is slower and begins already in the backswing phase. The last movements that are added to the hitting movement are those in the wrist (as the furthest joint from the central axis): radial adduction, supination, and extension (at the end of the hitting phase).

The analysis of the cycle movements in different joints, changes in the direction of these movements, and the timing of their occurrence leads to the conclusion that the coordination of the topspin backhand stroke is similar in most body segments.

The course of movements in the analyzed joints was accompanied by a large (*p* < 0.01) variability between the two groups determined using the ANOVA SPM test. Perhaps this was also caused by the sizable intragroup differentiation in the groups of men and women, a phenomenon proved by the high SD values and emphasized in our previous studies [26,37]. The differences revealed by the SPM test are likely also related to a different time of occurrence of events and duration of individual phases. In women, the backswing phase started earlier and was longer than in men, which caused a shift at the beginning of the hitting phase. Similarities in the coordination of movement are observed, for example, in the joints of the lower limbs. In the knee joints, despite generally similar movement directions in the individual phases, the SPM test showed no differences between the groups only in about half of the back to ready position phase and at the beginning of the backswing phase. The smallest number of periods with different patterns can be observed during the flexion/extension movement in the hip joints, especially on the non-playing side. There was flexion in the backswing phase in both groups. Next, before the beginning of the hitting phase, the extension movement starts; this is a period without intergroup differences (from ca. 55% to 95% of the hitting cycle) on the non-playing side. The differences are significant in the hip joint on the playing side in this part of the cycle (hitting phase), although the directions of movement are similar. This effect is probably due to the higher speed and slightly higher range of this movement in men. Differences can also be observed in abduction, although the directions of movements are similar in both groups, and the change in direction occurs at similar moments of the cycle.

A similar pattern can be noticed for movements in the shoulder joint (flexion and abduction). Major intergroup differences, also confirmed by the SPM test, were observed in the movement of thoracic rotation (a higher range of motion and speed in men at the end of the backswing phase and during most of the hitting phase) and the use of shoulder external rotation in the hitting phase by men. Apparent differences were also detected in the elbow joint, with women extending their limbs throughout the backswing and hitting phases, using a higher range of this movement than men. Flexion and rapid extension are also observed during the hitting phase in this joint. The higher speed and larger range of movements in the extension and adduction in the wrist in men and the supination in this joint in women were also notable. Perhaps these differences result from the variability of the anatomical structure, different functionalities of body parts etc. [38,39]. However, these differences may also be due to tactical requirements, especially in male table tennis. It is noticeable that in table tennis at the world elite level, topspin backhand is not only the first attack but also a stroke that directly yields a point [5]. Quantitative analyses show that men attack more often, use higher forces, and take more risk than women [40]. The use of upper body rotation in men, with the additional dynamic extension in the elbow and wrist joints and the external rotation of the shoulder during the hitting phase, is likely to result in a higher acceleration of the playing hand in male players. The aforementioned upper limb movements have been indicated as the main movements supporting the increase in the racket speed during topspin backhand stroke in table tennis [6]. The bigger range of supination in wrist in females than in males during hitting is probably the manifestation of a compensation mechanism.

Many studies concerning various sports have found a decrease in the differences between the kinematic parameters of elite athletes as the critical moment—an accurate strike, throw, etc.—was approached (related to the accurate result of the movement) [16,41,42,43]. This thesis is reflected in the occurrence of periods without differences in the results of the SPM test during the hitting phase, especially in the second part, before reaching ACCMax. Such periods were found for wrist movements and hip flexion. Perhaps these elements are common to both groups in coordinating the movement in the phase that is critical during a topspin backhand stroke. The main limitation of our research is the size of both groups. Therefore, our conclusions should be approached with caution. Despite the small group size, there was quite a large variation in the course of movements in the individual joints, which makes it challenging to interpret the results.

## 5. Conclusions

In the present study, we analyzed the coordination of the topspin backhand stroke against the backspin ball in elite female and male table tennis players. The proximal-to-distal sequencing in the movement pattern was identified in both groups in the moments of change in direction and speed of movements in individual phases. The coordination of movements pattern during the topspin backhand shot presented in our study may form a guideline for coaches and help to understand the principle of proximal-to-distal sequencing in table tennis backhand shots. The application of the SPM method allowed us to find many gender differences in movement patterns in the selected joints. A first example comprises the differences in the contribution of thoracic rotation, external shoulder rotation, and dorsal flexion in wrist during the hitting phase (backswing–forward) as evidenced by the bigger movement range in males than females during this phase. The use of upper body rotation in men, with the additional dynamic extension in the elbow and wrist joints and the external rotation of the shoulder during the hitting phase, is likely to result in a higher acceleration of the playing hand. The bigger range of supination in wrist in females than in males during hitting is probably the manifestation of a compensation mechanism. Perhaps these differences result from the anatomical gender differences in body build (e.g., mass of the body segments) or other functionalities of individual body parts in both groups. However, they are also likely to result from tactical requirements, especially the need for a stronger attack in men. The gender differences presented in this study can help in the individualization of the training process in table tennis.

The occurrence of periods without differences in test results during the hitting phase, especially in the second part before reaching ACCMax, was also demonstrated. Such periods were found for wrist movements and hip flexion. Perhaps these elements are common to both groups in coordinating the movement in the phase that is critical during a topspin backhand shot.

## Figures and Tables

**Figure 1 ijerph-17-06996-f001:**
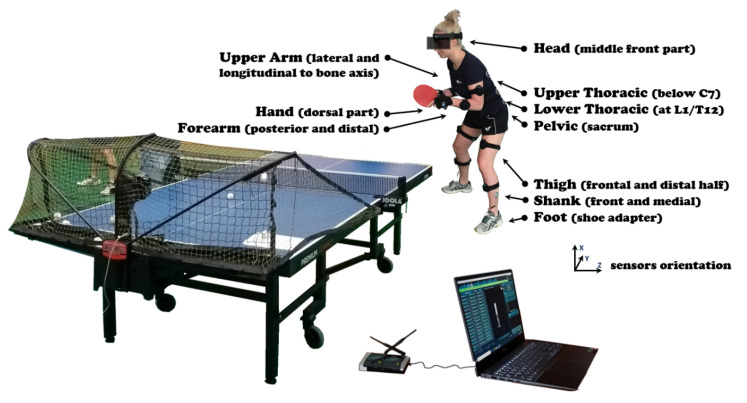
The measurement setup with the computer workstation, locations of the sensors on the player’s body.

**Figure 2 ijerph-17-06996-f002:**
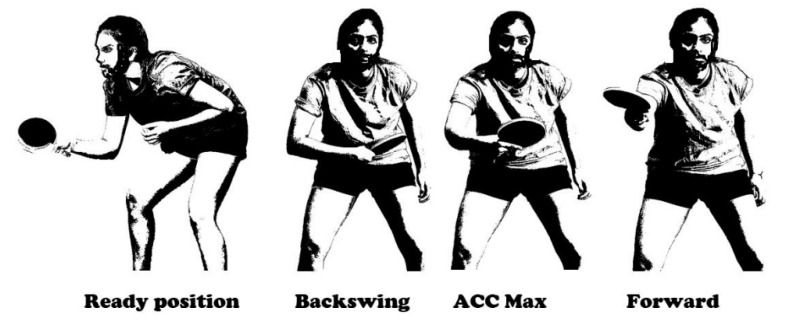
The indicated events of topspin backhand stroke.

**Figure 3 ijerph-17-06996-f003:**
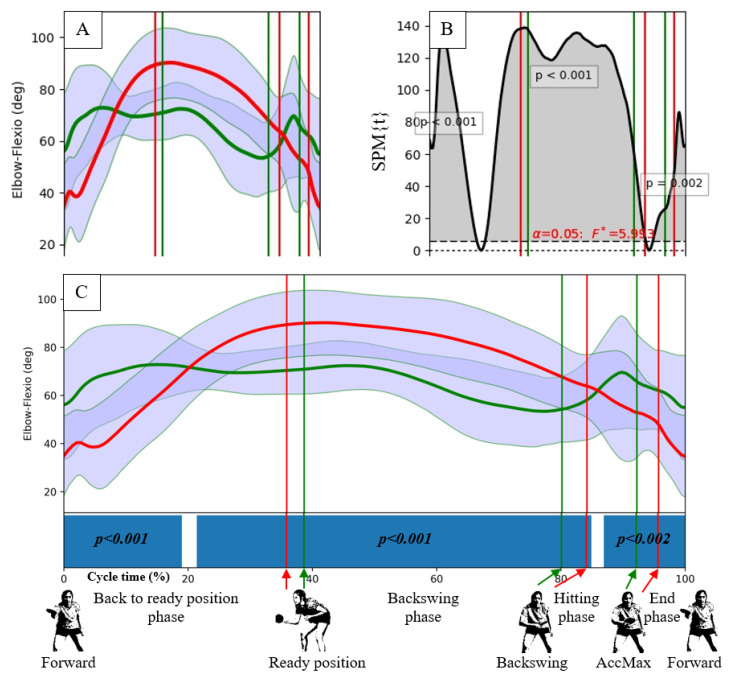
The statistical parametric mapping (SPM) procedure. For each data set (**A**) sampled every 0.1% of cycle time, an SPM{t} function (**B**) was calculated using Python. The SPM statistical information was transformed into a horizontal bar representation (**C**), designating the most significant differences between the compared patterns. The green curve presents the data for men, whereas the red one for women. The vertical lines represent the successive movement positions for men (green) or women (red) described with still images at the bottom.

**Figure 4 ijerph-17-06996-f004:**
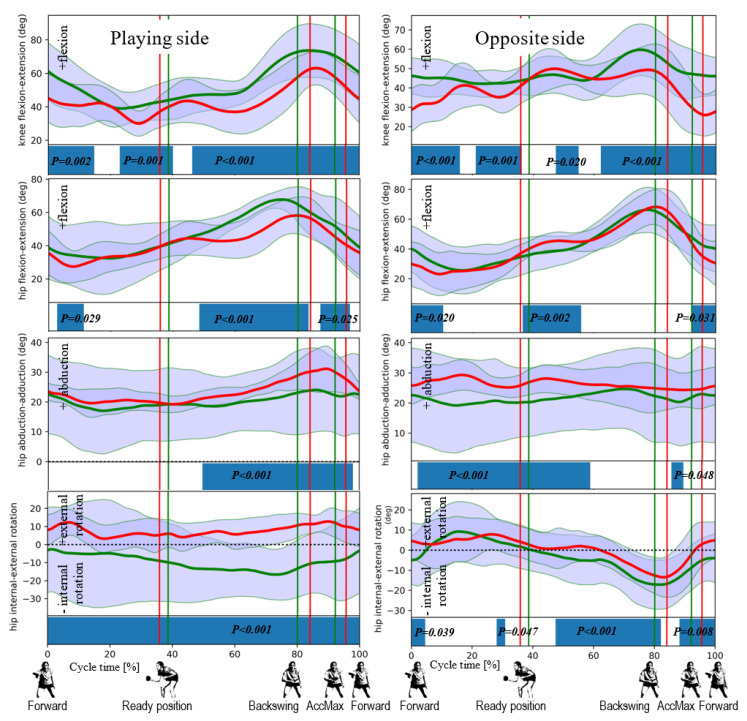
Differences in low-extremity angular kinematics between female (in red) and male (in green) players revealed by the SPM for knee flexion-extension, hip flexion-extension, hip abduction-adduction, and hip internal-external rotation for the playing side (**left column**) and opposite/contralateral side (**right column**). The statistically significant differences (for *p* < 0.05) are marked on the horizontal bar.

**Figure 5 ijerph-17-06996-f005:**
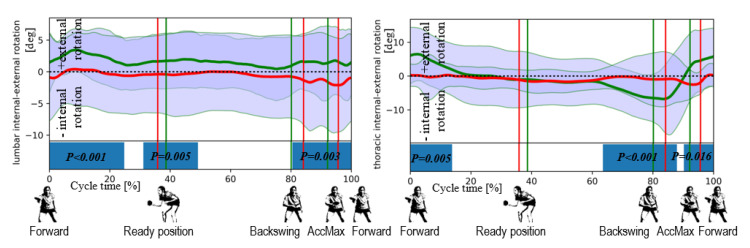
Differences in torso angular kinematics between female (in red) and male (in green) players revealed by the SPM for the lumbar internal-external rotation (**left**) and thoracic internal-external rotation (**right**). The statistically significant differences (for *p* < 0.05) are marked on the horizontal bar.

**Figure 6 ijerph-17-06996-f006:**
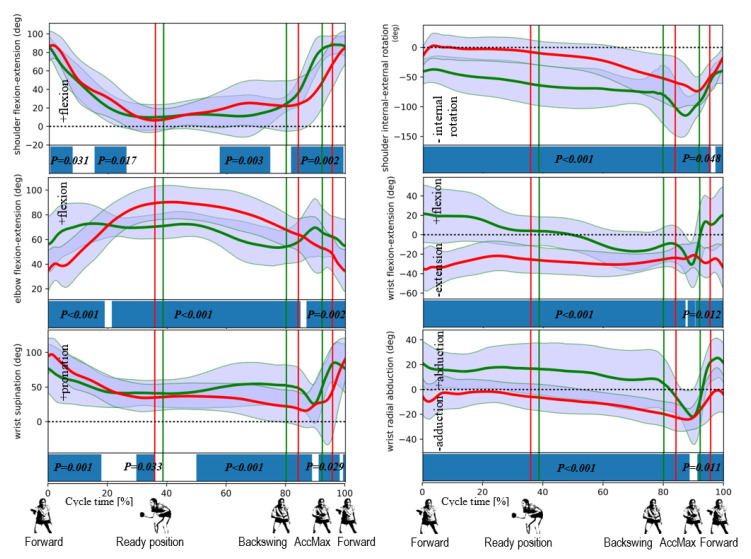
Differences in upper body angular kinematics between female (in red) and male (in green) players revealed by the SPM for the shoulder flexion-extension, shoulder abduction-adduction, shoulder internal-external rotation, elbow flexion-extension, wrist flexion-extension, wrist supination, and wrist radial abduction for the playing side. The statistically significant differences (for *p* < 0.05) are marked on the horizontal bar.

**Table 1 ijerph-17-06996-t001:** The moment when cycle events occur in the test groups (in %, mean ± standard deviation (SD).

Variable	Ready	Backswing	ACCMax	Forward
**Female**	35.82 ± 14.83	84.10 ± 3.44	86.19 ± 1.37	95.60 ± 3.19
**Male**	38.62 ± 14.30	80.07 ± 5.36	89.77 ± 2.18	92.11 ± 3.36
***p* of Wilcoxon test**	0.08	<0.01 ٭	<0.01 ٭	<0.01 ٭

٭-the singnificannt differences.

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
