# Peer review of "Statistical Parametric Mapping Reveals Subtle Gender Differences in Angular Movements in Table Tennis Topspin Backhand"

_ijerph, 2020, doi:10.3390/ijerph17196996_

Round 1
Reviewer 1 Report
Please see the attachment.

Author Response
Responses to the Editor's and Reviewers' Comments.
Dear Editors & Reviewers of the International Journal of Environmental Research and Public Health,
we appreciate very much for all the constructive comments and useful observation. We also thank You for the effort and time put into the review of our manuscript.
In following response each comment has been carefully considered point by point and replied. Responses to the reviewers are in the form of a dialogue and marked with tab and italics; changes in the revised manuscript are marked by “track changes” function of MS Word
RESPONSE TO REVIEWER NO.1
Overall comments:
- This paper conducts interesting research on how to quantify the body’s motion for the
tennis players. Their methods to conduct the experiment and post-processing data needs
more clarification and improvement.
Abstract:
Please clearly mention the original contribution the paper to this field
- We added two sentences in the Abstract about original contribution of our work to the field of movement analysis: in ll. 24-26 It reads: “The study presents movement patterns in topspin backhand shot, describing the proximal-to-distal sequencing principle during this shot.” and in ll. “Presented in this study gender differences can help in the individualization of the training process in table tennis.”
Pg1-L23-25: Please revise the sentence, it is vague and has grammar issues
- Thank you very much for this comment. We rewrote this sentence. It stays now: “. The SPM study revealed differences between men and women in the contribution of thoracic rotation, external shoulder rotation, dorsal flexion, and supination in the wrist during the hitting phase.”. And we also used professional proofreading service.
Introduction:
Please split the introduction to several paragraphs, I’ve never seen one long paragraph like this for
introduction. Split it to works that have been done on: 1-Kinematics quantification, 2- available
theories, available statistical methods, and etc. The last paragraph can shortly explain what you
have done and summary of results.
- Thank you for the useful observation. We split the Introduction into four subsections: 1.1. Table tennis kinematics in scientific literature; 1.2 Variability—an interesting issue in table tennis; 1.3 Statistical Parametric Mapping and 1.4 The aim of the research and motivation.
Pg2-L45, Close the bracket. point (e.g. [3-6]. Authors
- Thank you. It has been corrected.
Materials and Methods
I find that the author needs to improve this section by addressing important of areas that have not
been looked in this study. First, finding the patterns of players kinematics without considering the output of each strokes can be seriously biased. Only considering the marked area to hit by players without considering the acceleration/speed of the ball, racket kinematics, ball speed and hall height can result change the pattern. By taking video of each test and doing some simple image processing the output data can be more efficient.
- Thank you for the useful observation. The output of each strokes was considered prior to analysis. In our experiment we tried to set the external factors influencing measured pattern fixed as much as possible. For example, we used a dedicated robot to control the ball pitch (its direction, flight trajectory and landing area) and speed. We indeed took the video of every task in further analysis, because, as we wrote in the Methods section: “only successful shots considered “on table” and played diagonally were recorded for further calculations (missed balls, balls hit out of bounds, and balls hit into the net were excluded)” we considered for further analysis only positive output of the task - diagonally hit on table. To make it clear we added: “After video analysis only successful shot....”
Figure 1: The location of the sensors and direction of axis are not very clear. You can diffintly show
then in a body scheme with the axis shown on them. Something like this:
- The locations on the Figure were adjusted in addition to the detailed description in the article text. The orientation of each segment is considered in the software of our system and is in accordance with the International Society of Biomechanics (ISB) recommendations on definitions of joint coordinate systems of various joints for the reporting of human joint motion (we gave a proper citation). In the motion analysis, Euler rotational angles (pitch, yaw, roll) are rather used then Cartesian coordinates. Only acceleration data is in sensor coordinates, meaning the acceleration directions are relative to the x‐, y‐ and z‐ axes of the calibrated sensor. The axes directions are based on the position the sensors are during calibration. We agreed that the specific direction were vague, that’s why, in addition, we chose to supplement the resulting graphs with +/- directions, for better clarity.
The author did not mention about how the data were filtered and calibrated to use for building
the statistical model. This part is one of the most critical part of your research that is not well
explained in method.
- Noraxon’s IMU technology mathematically combines and filters incoming source signals on the sensor level and transmits the 4 quaternions of each sensor. We used system-build fusion algorithms and Kalman filtering (digital bandpass Finite Impulse Response filter (FIR)). But exact parameters of the filters are known only to the manufacturer. We also used “magnetic stabilization” for better magnetometer-based drift correction, which is advised by Noraxon. This mode allowed directly access to all unprocessed IMU sensor data. We added the above information in ll. 174-177.
Also, did you use the pitch, yaw, and roll angel and rate directly or you transferred them to global
coordinates. You can also show the joint coordinate systems using a figure rather than
explaining them one by one.
- Firstly, sensors were calibrated according to the factory recommendations. We added this information to the manuscript. The ongoing use of MyoMOTION requires a monthly reset of the Gyro sensors (“Zero Gyro” procedure). While calibrating the sensors are not moved for approx. 1-2 minutes. Secondly, prior taking a measurement the IMU sensors and avatar start in calibration mode (visible on preview screen) and as such will not provide data until after calibration. The sensors are compared to a known starting position (calibration type) of all the body joints. The body segments assigned to the sensors during configuration are represented by a skeletal avatar. While calibrating, the subject stays in upright standing position with neutral (=0 deg) joint position and is not to move for approx. 1-2 minutes. All the calibration procedures are semiautomated and implemented in the MyoMotion software. The joint coordinate system is only a step in the Direct Linear Transformation (DLT) procedure, which transfers sensors (and joint) coordinates into one 3D global coordinate system. In the description of the results we used resulting medical angles instead, that’s why the definition of these angles is so crucial. We also added information in the text: “Every participant, at the beginning of the measure was checked and the system was calibrated according to the factory recommendations”.
Information about the participants were only the age, height, and body mass. Don’t you think
there might be other physical factors that affect the players kinematics and you can include in your
study like years of experience that they have. Please explain about that as well. Also, did you find
any of age, height, body mass as an important factor in kinematic responses of players
- We provided the information about players’ years of experience. Unfortunately, we didn't take any anthropometric measures of the players, but we must admit, that these (as well as the relation between height, body mass or age and kinematics) would be also an interesting issue that could also be taken into account.
Please separate the results from the method section in your paper.
- These chapters have been separated.
The way that you defined the cycle is very subjective. For example, how do you define the impact
point as the point that reach the maximum acceleration. Should this be the maximum velocity of
the racket?
Please clearly explain the cycle that you considered in you study to show the results and do the
statistical analysis.
- Thank you very much for this doubt and question. For the further analysis we divided cycle into four phases (Back to ready position, Backswing, Hitting phase and End Phase) and we defined four events: Forward (as an end of the previous cycle and beginning of next one), Ready position, Backswing and ACC Max. ACC Max is the moment, when the hand reaches the maximum acceleration during the forward movement. We didn’t measure the kinematics of the racket and, also, we didn’t consider the ACC Max moment as a moment of racket’s contact with the ball. It is in accordance to our previous study (), where we compared the two instances. To make it clearer we provided the figure with presentation of the events.
Conclusion/Discussion
In discussion you mentioned about the speed in Page 10-line 334, I did not see that in your data
Analysis.
- Thank you very much for this observation. The speed (or velocity) was not, indeed, directly measured by us. Indirect observation allowed us to use the term speed as a magnitude in the angular range of change in the specific time period (or phase). To make it clearer we explained in l. 371 “(i.e. the range of movement in the phase)”.
Please explain how the results of your study (finding a pattern) contribute the body of knowledge
in this area and which parameters are important in your view.
- To emphasize our findings, we added in Conclusions chapter: ll. 445-448 - The coordination of movements pattern during the topspin backhand shot presented in our study may form a guideline for coaches and help to understand the principle of proximal-to-distal sequencing in table tennis backhand shots. The application of the SPM method allowed for finding many gender differences in movement patterns in the selected joints” and ll. 456-459: “The gender differences presented in this study can help in the individualization of the training process in table tennis”.
Reviewer 2 Report
The manuscript were dedicated to study of gender differences in biomechanic movements in table tennis. But in conclusion gender differences are not sufficiently shown. What the reasons of differences with parameters of movements between female and male athletes: antropometric, movement techniques or personal?
Author Response
Responses to the Editor's and Reviewers' Comments.
Dear Editors & Reviewers of the International Journal of Environmental Research and Public Health,
we appreciate very much for all the constructive comments and useful observation. We also thank You for the effort and time put into the review of our manuscript.
In following response each comment has been carefully considered point by point and replied. Responses to the reviewers are in the form of a dialogue and marked with tab and italics; changes in the revised manuscript are marked by “track changes” function of MS Word
RESPONSE TO REVIEWER NO.2
The manuscript were dedicated to study of gender differences in biomechanic movements in table tennis. But in conclusion gender differences are not sufficiently shown. What the reasons of differences with parameters of movements between female and male athletes: antropometric, movement techniques or personal?
- Thank you very much for your question and comment. We have supplemented Conclusions changing few sentences: ll. 447-455 - “The application of the SPM method allowed for finding many gender differences in movement patterns in the selected joints. A first example comprises the differences in the contribution of thoracic rotation, external shoulder rotation, and dorsal flexion in wrist during the hitting phase (backswing–forward) as evidenced by the bigger movement range in males than females during this phase. The use of upper body rotation in men, with the additional dynamic extension in the elbow and wrist joints and the external rotation of the shoulder during the hitting phase, is likely to result in a higher acceleration of the playing hand. The bigger range of supination in wrist in females than in males during hitting is probably the manifestation of a compensation mechanism. “ and “Perhaps these differences result from the anatomical gender differences in body build (e.g., mass of the body segments) or other functionalities of individual body parts in both groups. However, they are also likely to result from tactical requirements, especially the need for a stronger attack in men. The gender differences presented in this study can help in the individualization of the training process in table tennis”.
Reviewer 3 Report
Dear Authors,
Thanks for giving me the chance to read this manuscript, “Statistical Parametric Mapping reveals subtle gender differences in angular movements in table tennis topspin backhand”. the patterns of movement of topspin backhand stroke in table tennis and to use SPM to compare these patterns between advanced female and male players are increasingly attracting academic attention. The current paper tries to reproduce the pattern of movement in the joints during topspin backhand strokes in the studied athletes.
Generally, it is a timely, significant, and interesting topic, and the paper looks structured with clear direction and research questions. However, there are several serious points worth noting which must be appropriately addressed.
- A more explicate justification of the difference between SPM and other methods
A detailed comparison of the proposed method and SPM should be described to help readers get a further understanding of this method.
- When should we use SPM?
- Why should I use SPM rather than other time-series methods, such as moving average, exponential smoothing, stationarity, autocorrelation, or even SARIMA?
- What should we be concerned about when introduced this method?
- What might be the limitation of this method?
- Is there any code in Github for further validation?
Reference:
Wei, W. W. (2006). Time series analysis. In The Oxford Handbook of Quantitative Methods in Psychology: Vol. 2.
- Format issue
In-Line 348, Table 1. seems does not have the correct format according to the author guideline. Similar problems could be seen in many places, such as Line 278. And there are many typos in this manuscript. The authors are advised to ask a third party to proofread and carefully format the manuscript for further consideration.
To sum up, I personally like this paper and its contributions, and it is definitely a potentially publishable paper. However, the issues mentioned above should be carefully addressed in order to further proceed. Hope these suggestions help.
Author Response
Responses to the Editor's and Reviewers' Comments.
Dear Editors & Reviewers of the International Journal of Environmental Research and Public Health,
we appreciate very much for all the constructive comments and useful observation. We also thank You for the effort and time put into the review of our manuscript.
In following response each comment has been carefully considered point by point and replied. Responses to the reviewers are in the form of a dialogue and marked with tab and italics; changes in the revised manuscript are marked by “track changes” function of MS Word
RESPONSE TO REVIEWER NO.3
Dear Authors,
Thanks for giving me the chance to read this manuscript, “Statistical Parametric Mapping reveals subtle gender differences in angular movements in table tennis topspin backhand”. the patterns of movement of topspin backhand stroke in table tennis and to use SPM to compare these patterns between advanced female and male players are increasingly attracting academic attention. The current paper tries to reproduce the pattern of movement in the joints during topspin backhand strokes in the studied athletes.
Generally, it is a timely, significant, and interesting topic, and the paper looks structured with clear direction and research questions. However, there are several serious points worth noting which must be appropriately addressed.
- A more explicate justification of the difference between SPM and other methods
A detailed comparison of the proposed method and SPM should be described to help readers get a further understanding of this method.
- When should we use SPM?
- Why should I use SPM rather than other time-series methods, such as moving average, exponential smoothing, stationarity, autocorrelation, or even SARIMA?
- What should we be concerned about when introduced this method?
- What might be the limitation of this method?
- Thank you very much for your questions and comments. We provided more information about the SPM method, with new references as well, which form answers to your questions. In ll. 95-112 it stays now: “Recent advances in SPM for continuum one-dimensional data (e.g., kinematic or kinetic time series) have been adopted by the research community—e.g., scientists, clinicians, engineers, and all individuals who deal with spatiotemporally continuous data. Originally adopted for the analysis of pedobarographic images [21], SPM has also been useful in uni- or multivariate time series analysis (kinematics, kinetics, or surface EMG data) [19, 20, 22]. Recently, a classical SPM supplemented with a Bayesian version was proved to be more sensitive to assess significant side differences, is not asymmetric like the classical version, and takes an alternative hypothesis into account [23]. The SPM is especially suitable for biomechanical data that is spatiotemporally smooth, sampled above the Nyquist frequency, and bounded in time or space [24]. The two main advantages of SPM over other numerical approaches (such as the autocorrelation/autocovariance function, moving average or autoregressive representations of time series processes, exponential smoothing, Fourier analysis, or time/vector time series analysis) are: (1) the statistical results are presented directly in the original sampling space, so their spatiotemporal biomechanical framework is immediately apparent and (2) there is no need for assumptions regarding the spatiotemporal central points of signals [20]. As other methods, SPM also has some known limitations: e.g., it requires temporal data normalization, non-random sampling, and non-blind experimentation; its procedures are more complex—the random-field-theory-based inference especially; and it presents restricted accessibility (to Python/Matlab programming)”.
- Is there any code in Github for further validation?
- Yes: https://github.com/0todd0000
Reference:
Wei, W. W. (2006). Time series analysis. In The Oxford Handbook of Quantitative Methods in Psychology: Vol. 2.
- This reference only to the general - spatial use of SPM for medical application developed originally by Friston. SPM 1D is a simplified model which is implemented in the Python calculation algorithm (www.spm1d.org).
- Format issue
In-Line 348, Table 1. seems does not have the correct format according to the author guideline. Similar problems could be seen in many places, such as Line 278. And there are many typos in this manuscript. The authors are advised to ask a third party to proofread and carefully format the manuscript for further consideration.
- We sent our manuscript to English editing service, to form it accordingly with the Instructions for authors
To sum up, I personally like this paper and its contributions, and it is definitely a potentially publishable paper. However, the issues mentioned above should be carefully addressed in order to further proceed. Hope these suggestions help.
- Thank you very much for your assessment
Round 2
Reviewer 1 Report
Hi,
Thanks for addressing the comments and explanations. please see my comments in the attached file.
For further review, PLEASE send the latest clean version with accepted changes of the manuscript. It was very difficult to track how you address the comments in the text.
Best,
Reza

Reviewer 3 Report
The authors have successfully addressed my concern. I am happy to recommend this paper published in our journal.
Good job!
Author Response
Thank you very much for your opinion